

# Localization engineering by resonant driving in dissipative polariton arrays

**Gonzalo Usaj**

Centro Atómico Bariloche and Instituto Balseiro, Comisión Nacional de Energía Atómica (CNEA)- Universidad Nacional de Cuyo (UNCUYO), 8400 Bariloche, Argentina
Instituto de Nanociencia y Nanotecnología (INN-Bariloche), Consejo Nacional de Investigaciones Científicas y Técnicas (CONICET), Argentina
CENOLI, Université Libre de Bruxelles - CP 231, Campus Plaine, B-1050 Brussels, Belgium

usaj@cab.cnea.gov.ar

## Abstract

Arrays of microcavity polaritons are very versatile systems that allow for broad possibilities for the engineering of multi-orbital lattice geometries using different state preparation schemes. One of these schemes, spatially modulated resonant driving, can be used, for instance, to selectively localize the polariton field within the particular region of the lattice enclosed by the driving laser. Both the frequency and the spatial amplitude distribution (module and phase) of the driven laser field are important and serve as a knob to control the leakage outside that region and hence the extend of the spatial localization. Here, we analyse both the linear and nonlinear regimes using the lattice Green function formalism that is particularly suitable for the case of polariton arrays described in a tight-binding approximation. We identify the conditions for the laser induced localization to occur on arbitrary lattice's geometries and discuss some experimentally relevant cases. We find that the polariton-polariton interaction leads to a frequency shift of the optimal localization condition that could be used to further control it.

# 1 Introduction

Microcavity polaritons, composed quasiparticles formed by the coherent coupling of quantum well excitons and cavity photons, offer an unique opportunity for the study of driven-dissipative interacting bosonic systems [1, 2]. One particularity of these systems, in contrast to other bosonic systems in condensed matter, is that they are inherently out of equilibrium as they need to be externally pumped, either resonantly or non-resonantly, in order for the polaritons population to be created on the first place. Due to that, it is also possible to control the relevance of the interactions as the system can be taken from a linear regime (low power excitation), where interactions can be neglected, to a nonlinear regime (high power excitation) where the polariton-polariton interaction and/or the polariton-reservoir interactions (in the non-resonant case) are important and lead to several interesting non-trivial effects. This versatility, together with the available experimental accessibility and design capabilities has allowed the observation of many interesting phenomena such as superfluidity [3, 4], lasing from edge states [5], topological effects [6, 7], non-trivial spin effects [8], vortex quantization [9], optomechanical effects [10–13], or the recent observation of Kardar–Parisi–Zhang universal behaviour of the polariton field phase [14], among many others [15–19]. All these advances required an adequate tailoring of the polariton's arrays, such as its geometry or multi-orbital character, as well as the use of different techniques to control the polariton dynamics, such as properly engineered non-resonant reservoirs or suitable resonant driving schemes. In this work we will focus on the later case.

Properly designed patterns of resonant driving lasers have been used recently to create specific configurations of localized polaritons in 1D and 2D arrays [20–22] where the polariton field is essentially confined to the region enclosed by the driving lasers. This has been done both in the linear and nonlinear regimes. In the latter case the formation of an in-gap soliton in a Su–Schrieffer–Heeger (SSH) chain was also reported [20]. The results were interpreted by comparison with the numerical solution of a generalized Gross-Pitaeskii equation (gGPE) [23].

In this work we discuss a related alternative approach based on lattice Green functions that allows for a generalization to the case of arbitrary lattices and driving patterns. Using this we are able to determine the optimal conditions for the localization (confinement) of the polariton field inside the region enclosed by the driving lasers in a generic scenario in any dimension, hence providing a simple description of the experimental results of Ref. [21] while at the same time offering a conceptual basis for the design of future experiments. Our approach also allows, under certain conditions, to include the nonlinear effects mentioned earlier. We apply it to $1D$ arrays and present both numerical and analytical results that illustrate the basic underlying phenomena for localization and the role of the interactions. We find that the polariton-polariton interaction introduces a (mode dependent) frequency shift of the localization condition that can be tuned with the amplitude of the driving [24]. In the particular case of the SSH chain, we provide a clear description for the formation of the in-gap soliton and its hysteric behavior [20].

# 2 Tight-binding model for polaritons

We consider an arbitrary array of polariton cavities. For simplicity, only a single mode is considered on each cavity—a generalization to many modes/orbitals is straightforward. Assuming that the modes of each cavity are strongly bound, one can construct from the gGPE a simplified tight binding Hamiltonian $H$, as it is usually done for Bloch electrons. We only consider the coupling between nearest neighbor cavities, and ignore the effect of non-orthogonality between different cavity modes [25]. We restrict ourselves to the semi-classical approximation,

where the bosonic operators are replaced by complex functions, which implies that the solution of interest contains a large number of polaritons so quantum fluctuation can be ignored. The dynamics of the system is then given by the following set of coupled nonlinear equations,

$$i\hbar \frac{d\phi_j}{dt} = \sum_{k=1}^{M} H_{jk}\phi_k - i\frac{\hbar\gamma}{2}\phi_j + \hbar F_j e^{-i\omega_d t}, \tag{1}$$

with $j = 1 \ldots M$, $M$ being the total number of sites on the lattice (which could be finite or not). Here we added the term $\hbar F_j e^{-i\omega_d t}$ to describe the resonant driving, with frequency $\omega_d$, and included the losses on each site, characterized by the rate $\gamma$ (different rates could also be considered). In a mean field approximation, intra-cavity polariton-polariton interactions can be included by replacing $\hbar\omega_0 \rightarrow \hbar\omega_0 + \hbar U|\phi_j|^2$, where $\hbar\omega_0 = H_{jj}$ is the bare energy of the cavity mode and $\hbar U$ the polariton-polariton interaction strength. In the following sections we analyse the stationary solutions of Eq. (1) for different experimentally relevant cases using the lattice Green function of the system both for $U = 0$ and $U \neq 0$.

## 3 Linear regime

Let us first consider the case where the driving power (proportional to $|F_j|^2$) is small so that the nonlinear terms can be safely ignored ($U = 0$). Since we look for stationary solutions, all the modes must evolve at the frequency of the drive. Then, by moving to the rotating frame of the drive, $\phi_j(t) = \bar{\phi}_j e^{-i\omega_d t}$, we get

$$\hbar\omega_d \bar{\phi}_j = \sum_{k=1}^{M} H_{jk}\bar{\phi}_k - i\frac{\hbar\gamma}{2}\bar{\phi}_j + \hbar F_j, \tag{2}$$

whose general solution is

$$\bar{\phi} = G(\tilde{\omega}_d) \cdot F, \tag{3}$$

with $\bar{\phi} = (\bar{\phi}_1, \bar{\phi}_2, \ldots, \bar{\phi}_M)$, $F = (F_1, F_2, \ldots, F_M)$, and $\tilde{\omega}_d = \omega_d + i\gamma/2$. Here $G(\omega) = (\omega I - H)^{-1}$ is the Green function of the tight-binding lattice described by $(H)_{ij} = H_{ij}/\hbar$. Since there are numerous numerical methods to obtain $G(\omega)$ for any lattice or arbitrary array of cavities, it is rather straightforward to get $\bar{\phi}$ for any driving profile. Furthermore, in some specially simple cases even analytical solutions are possible. We note in passing that a similar approach was used in Ref. [26] to describe bound-states-in-the-continuum and its relation to localization in photonic lattices.

In addition, Eq. (3) leads to a general sum rule satisfied by the sum of the amplitudes of the modes, $I = \sum_j |\bar{\phi}_j|^2$, that reflects the balance between driving and dissipation. Namely,

$$
\begin{aligned}
I &= \sum_{j,\alpha,\beta} (G_{j\alpha}(\tilde{\omega}_d))^* F_\alpha^* G_{j\beta}(\tilde{\omega}_d) F_\beta, \\
&= \left(\frac{1}{\gamma}\right) F^* \cdot A(\tilde{\omega}_d) \cdot F, \tag{4}
\end{aligned}
$$

where we have introduced the spectral function, $A(\tilde{\omega}_d) = i(G(\tilde{\omega}_d) - G(\tilde{\omega}_d)^\dagger)$, and used that it satisfies $A(\tilde{\omega}_d) = \gamma G(\tilde{\omega}_d)^\dagger \cdot G(\tilde{\omega}_d)$. Note that $A_{jj}(\omega) = -2\text{Im}(G_{jj}(\omega))$ relates directly to the local density of states $D_j(\omega) = -(1/\pi)\text{Im}(G_{jj}(\omega))$.

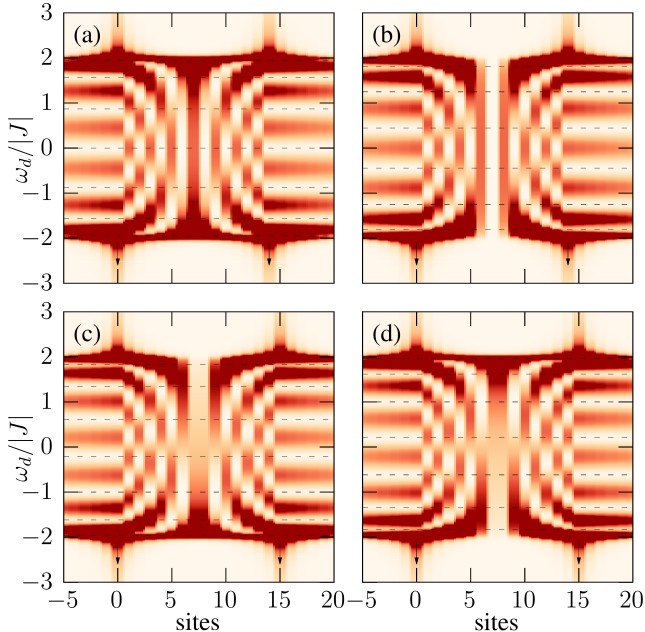

Figure 1: Colormap of $|\phi_j|^2$ as a function of the driving frequency for the case of a linear array with two driven sites (indicated by the black arrows): (a) and (b) correspond to $F_{14} = F_0$ (even excitation) and $F_{14} = -F_0$ (odd excitation), respectively, while (c) and (d) to $F_{15} = F_0$ and $F_{15} = -F_0$. The horizontal dashed lines correspond to $\omega_d = \Omega_m = 2J\cos(m\pi/N)$, with $m = 1, \ldots, N-1$ and $N = 14, 15$. Here $\gamma = 0.05J$ and $\omega_0 = 0$. The color scale is saturated to enhance the contrast due to the presence of the band edge singularities. Dark (light) color reflects a large (small) amplitude.

## 3.1 Linear array

Let us now consider an infinite linear array with a nearest neighbors hopping described by the coupling $J = H_{j(j+1)}/\hbar = H_{(j+1)j}/\hbar$.

*Single site driving–*. We assume, without any loss of generality, that site 0 is the one being driven, $F_j = F\delta_{j0}$. Hence, $\bar{\phi}_j = FG_{j0}(\tilde{\omega}_d)$. Using the recursion relations satisfied by the Green function we can write, for an infinite linear array,

$$G_{j0}(\omega) = [Jg(\omega)]^{|j|}G_{00}(\omega), \tag{5}$$

where, for $|\omega - \omega_0| \leq 2|J|$, we have

$$
\begin{aligned}
g(\omega) &= \frac{(\omega - \omega_0) - \mathrm{i}\sqrt{4J^2 - (\omega - \omega_0)^2}}{2J^2}, \\
G_{00}(\omega) &= \frac{-\mathrm{i}}{\sqrt{4J^2 - (\omega - \omega_0)^2}}.
\end{aligned}
\tag{6}
$$

Here, $g(\omega)$ is the Green function of the surface site of a semi-infinite array, while $G_{00}(\omega)$ is the bulk Green function of an infinite array. Surface and bulk density of states are given by $-(1/\pi)\mathrm{Im}(g(\omega))$ and $-(1/\pi)\mathrm{Im}(G_{00}(\omega))$, respectively. Clearly, the amplitudes $|\bar{\phi}_j|^2 \propto e^{-\lambda|j|}$ decay exponentially with $\lambda = -\ln(|Jg(\tilde{\omega}_d)|)$, which is proportional $\gamma$.

*Two sites driving–*. We now consider the case where $F_j = F_0\delta_{j0} + F_N\delta_{N0}$, with $N \geq 2$. The solution is obtained from the previous case by linear superposition,

$$\bar{\phi}_j = G_{j0}(\tilde{\omega}_d)F_0 + G_{jN}(\tilde{\omega}_d)F_N. \tag{7}$$

These amplitudes are easily evaluated from Eqs. (5) and (6) and the translation invariance of the array. For the sites that are not in between the driven sites, we have

$$
\begin{aligned}
\bar{\phi}_{N+n} &= G_{(N+n)0}(\tilde{\omega}_d)F_0 + G_{(N+n)N}(\tilde{\omega}_d)F_N, \\
&= [Jg(\tilde{\omega}_d)]^n G_{00}(\tilde{\omega}_d)(F_0[Jg(\tilde{\omega}_d)]^N + F_N), \\
&= [Jg(\tilde{\omega}_d)]^n \bar{\phi}_N,
\end{aligned}
\tag{8}
$$

and

$$
\begin{aligned}
\bar{\phi}_{-n} &= G_{-n0}(\tilde{\omega}_d)F_0 + G_{-nN}(\tilde{\omega}_d)F_N, \\
&= [Jg(\tilde{\omega}_d)]^n G_{00}(\tilde{\omega}_d)(F_0 + F_N[Jg(\tilde{\omega}_d)]^N), \\
&= [Jg(\tilde{\omega}_d)]^n \bar{\phi}_0,
\end{aligned}
\tag{9}
$$

with $n \geq 0$. This shows, as before, that the amplitudes of the modes far from the pumped region are exponentially small as $|Jg(\tilde{\omega}_d)| < 1$ for $\gamma \neq 0$ regardless the value of $\omega_d$. It is evident that we could excite *only* the modes between the driven sites (maximum localization or confinement), so that $\bar{\phi}_{N+n} = \bar{\phi}_{-n} = 0 \ \forall n \geq 0$, if we can find a solution of

$$
F_0[Jg(\tilde{\omega}_d)]^N + F_N = F_0 + F_N[Jg(\tilde{\omega}_d)]^N = 0.
\tag{10}
$$

This equation implies that $[Jg(\tilde{\omega}_d)]^N = \pm 1$ and that $F_N = \mp F_0$. However, as mentioned above, because $\gamma \neq 0$ this condition cannot be strictly satisfied. Despite of that, the numerical solution obtained from Eq. (3) shows a significant localization of the polariton field within sites 0 and $N$ provided $\gamma/J$ is small (Fig. 1). Therefore, for the sake of argument for finding approximate conditions for an optimal localization, let us take $\gamma = 0$. In such a case, $Jg(\omega_d) = e^{i|k|}$ with $\omega_d = \omega_0 + 2J\cos k$ defining $k$. The condition $e^{i|k|N} = \pm 1$ requires that $k = m\pi/N$ with $m = 1, \ldots, N-1$—the values $m = 0$ and $m = N$ are excluded due to the divergence of $G_{00}(\omega)$ at the band edge. This has the clear interpretation that the stationary modes excited by the driving are the modes corresponding to the effectively 'finite' chain of $N - 1$ sites formed between the two driven cavities. By fixing the relative phase of the driving, $F_N = \mp F_0$, one can excite either even or odd modes if $\omega_d$ is chosen to satisfy $\omega_d = \Omega_m = \omega_0 + 2J\cos(m\pi/N)$ with the appropriate choice for $m$. Indeed, this result could have been anticipated from Eq. (10) as in such a case Eq. (1) for $\phi_j$ with $1 \leq j \leq N-1$ corresponds to the isolated chain.

Figure 1 shows a colormap of $|\bar{\phi}_j|^2$ obtained numerically from Eq. (3) for two different values of $N$: $N = 14$ (top panels) and $N = 15$ (bottom panels). Maximum localization occurs whenever $\omega_d \sim \Omega_m$ for some of the allowed values for $m$, indicated by the dashed horizontal lines, as expected (here we choose $\omega_0 = 0$). Interestingly, this corresponds to a minimum value of the intensity $I$, as can be directly confirmed from Eq. (4). This can also be understood if one notice that $I$ can be written as $I = (2/\gamma)\text{Im}(\boldsymbol{F} \cdot \bar{\boldsymbol{\phi}}^*)$. Then, since the maximum localization condition requires the amplitude of the driven sites to be small, of the order of $\gamma/J$, the factor $1/\gamma$ cancels out. Any other condition with a finite value on the driven sites (not proportional to $\gamma$) will lead to a higher value of $I$ [24, 26].

From the above considerations, it is clear that when $N = 2$, so that there is a single site ($j = 1$) between the driven cavities , the only possible solution that leads to the maximum localization is $\omega_d = \omega_0$ (corresponding to $k = \pm\pi/2$) and $F_2 = F_0$ as found in Ref. [21].

*Molecule*–. Another simple case corresponds to a finite chain of two sites, or a 'polariton molecule' [22]. A straightforward calculation gives for this system

$$
\boldsymbol{G}(\omega) = \frac{1}{\Delta\omega_1\Delta\omega_2 - J^2}\begin{pmatrix} \Delta\omega_2 & J \\ J & \Delta\omega_1 \end{pmatrix},
\tag{11}
$$

with $\Delta\omega_{1,2} = \omega - \omega_{1,2}$, and $\hbar\omega_1$ and $\hbar\omega_2$ the energy of site 1 and 2, respectively. Evidently, if we drive, say, site 1 at frequency $\omega_d = \omega_2$ then $\bar{\phi}_1 \sim (i\gamma/2J)F/J$ (assuming $|\omega_1 - \omega_2| \ll |J|$)

while the other site has $\bar{\phi}_2 \sim F/J$ and so the polaritons are localized on the undriven site, $|\bar{\phi}_1| \ll |\bar{\phi}_2|$, as observed in Ref. [22]. When both sites are driven it is possible to find a condition for one of the sites to have exactly zero amplitude. For instance $\bar{\phi}_1 = 0$ requires $F_1/F_2 = -J/(\omega_d - \omega_2 + i\gamma/2)$, that is, an specific amplitude/phase relation between the two driving lasers. In that case it is straightforward to verify that $\bar{\phi}_2 = -F_1/J$.

*Su–Schrieffer–Heeger (SSH) chain–.* Yet another interesting linear geometry is the case of an SSH chain (linear lattice with alternating hoppings $J$ and $J'$). Since Eq. (3) is valid for arbitrary lattices in any dimension (see next section), one only needs to specify $\mathbf{G}(\omega)$ for the SSH model. For simplicity, here we address only the situation where a single site (the 0 site) is driven with frequency $\omega_d = \omega_0$, which corresponds to a driving frequency inside the gap of the SHH chain—more details on this case are presented in the nonlinear section. It follows from simple considerations that both the real and the imaginary part of $G_{00}(\tilde{\omega}_d)$, neglecting corrections of order $\gamma/|J - J'| \ll 1$, are zero due to the chiral symmetry of the lattice and the fact that there is a gap at $\omega_0$—if one moves away from the center of the gap, the real part becomes different from zero while the imaginary part is always proportional to $\gamma$. Therefore, the driven site has nearly zero amplitude, $\bar{\phi}_0 = G_{00}(\tilde{\omega}_d)F \sim 0$. This implies that the 1D lattice is effectively split in two independent parts. As such, one of the sides has the proper termination to host an edge mode (so called zero energy mode, meaning energy $\hbar\omega_0$) while the other does not. Thus, only one side is excited with an amplitude profile decaying exponentially away from it (the decay length being determined by the inverse of the gap, under our assumption that $\gamma/|J - J'| \ll 1$), and with nonzero amplitudes (up to corrections of order $\gamma/|J - J'|$) only on a given sub-lattice. Indeed,

$$\bar{\phi}_{\pm j} = G_{\pm j,0}(\tilde{\omega}_d)F = J_\pm g_{\pm j,\pm 1}(\tilde{\omega}_d)\bar{\phi}_0\,, \tag{12}$$

where $j > 0$, $J_\pm$ is the hopping between site 0 and site $\pm 1$ (so it takes either the value $J$ or $J'$) and $g_{\pm j,\pm 1}(\omega)$ is the surface Green function connecting site $\pm 1$ and $\pm j$ of a semi-infinite SSH chain ending on site $\pm 1$. Because $\bar{\phi}_0 \sim 0$, or more precisely $\sim \gamma/|J - J'|$, in order for $\bar{\phi}_{\pm j}$ to have a finite value $g_{\pm j,\pm 1}(\omega)$ must have a pole at $\omega = \omega_0$. This is always the case for either $g_{j,1}(\omega)$ or $g_{-j,-1}(\omega)$ but not both so only one side of the chain is excited.

## 3.2 Arbitrary lattice and driving geometries

In this section we shall generalize the previous results to the case of an arbitrary lattice. For that, let us consider the case where $M$ sites of the lattice are driven. The array of these sites encloses a region $\mathcal{R}$, with $N - 1$ sites inside it, so that they separate $\mathcal{R}$ from the rest of the lattice, that is, each one of the sites in $\mathcal{R}$ are only connected to another site on $\mathcal{R}$ or to some of the driven sites. Note that it is not necessary for $\mathcal{R}$ to be a single connected region–it could be a ring shaped region or two separated ones [see Fig. (2)]. If we label the $M$ sites with Greek indices $\alpha$, $\beta$, ... and the rest of the sites outside $\mathcal{R}$ with $a$, $b$, ... it is then straightforward to show that $G_{a\beta}(\omega) = g_{ac}(\omega)V_{c\alpha}G_{\alpha\beta}(\omega)$—sum over repeated indices is assumed from hereon unless stated otherwise—where $g_{ac}(\omega)$ is the surface Green function of the external lattice sites surrounding the driven sites (ie. the one obtained by decoupling or removing the $M$ sites from the lattice) and $V_{c\alpha}$ ($\equiv H_{c\alpha}/\hbar$) the hopping matrix element connecting the $\alpha$-site (driven) with the $c$-site (external). Hence

$$\begin{aligned}\bar{\phi}_a &= G_{a\beta}(\tilde{\omega}_d)F_\beta\,, \\ &= g_{ac}(\tilde{\omega}_d)V_{c\alpha}\bar{\phi}_\alpha\,, \end{aligned} \tag{13}$$

where we used the fact that $\bar{\phi}_\alpha = G_{\alpha\beta}(\tilde{\omega}_d)F_\beta$. This shows that if, under the appropriate conditions described below, $|\bar{\phi}_\alpha| \sim (\gamma/|J|)|F/J| \ll 1$, where $|J|$ is some typical value for the

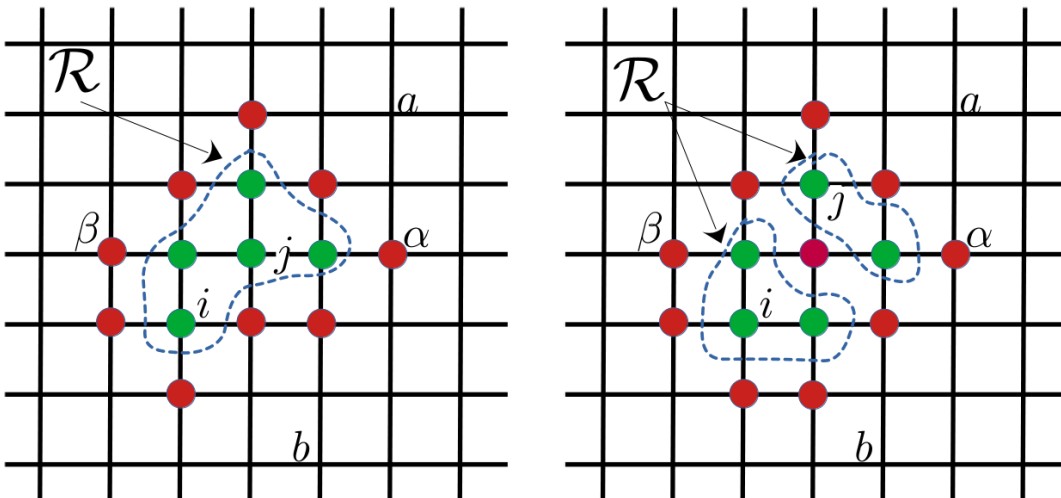

Figure 2: Scheme of different driving configuration on an arbitrary lattice. Driven sites are indicated by red dots and labeled by Greek letters ($\alpha$, $\beta$, ...). The region $\mathcal{R}$ enclosed by the driven sites is indicated by the green dots and labeled by $i$, $j$, $k$, .... Notice we do not require for it to be a single connected region nor a single one as far as it is surrounded by the driven sites. The rest of the lattice sites are labelled as $a$, $b$, $c$, ...

hopping matrix elements, then the external sites are also $|\bar{\phi}_a| \ll 1$ (further suppressed by the exponential decay of $g_{ac}(\tilde{\omega}_d)$ as a function of the distance $|r_a - r_c|$) induced by the finite linewidth $\gamma$ or a bulk gap, if present). Note here that in order to guarantee that $|\bar{\phi}_\alpha|$ is small it is important that $g_{ac}(\omega)$ has no singularities at $\omega = \omega_d$, which is usually the case unless there is a bound state at the surface (defined by removing the $M$ sites). The latter can happen only for special lattices and terminations or inside topological gaps hosting edge modes.

To calculate the amplitude $\bar{\phi}_j$ of the sites inside $\mathcal{R}$ it is helpful to change to the basis that diagonalizes the Hamiltonian $H_\mathcal{R}$ for the isolated region $\mathcal{R}$, $H_\mathcal{R} |\varphi_n\rangle = \hbar\Omega_n |\varphi_n\rangle$, so that $\bar{\phi}_n = \sum_{i \in \mathcal{R}} \langle\varphi_n|i\rangle \bar{\phi}_i$. Then

$$\begin{aligned}
\bar{\phi}_n &= G_{n\beta}(\tilde{\omega}_d) F_\beta, \\
&= g_{nn}^\mathcal{R}(\tilde{\omega}_d) V_{n\alpha} \bar{\phi}_\alpha,
\end{aligned} \tag{14}$$

where $\boldsymbol{g}^\mathcal{R}(\omega) = (\omega I - H_\mathcal{R})^{-1}$ is the Green function of the isolated region described by $(H_\mathcal{R})_{ij} = (H_\mathcal{R})_{ij}/\hbar = H_{ij}/\hbar$ with $i, j \in \mathcal{R}$ and $V_{n\alpha} = \sum_{i \in \mathcal{R}} \langle\varphi_n|i\rangle V_{i\alpha}$ is the hopping matrix element connecting $|\varphi_n\rangle$ to the $\alpha$-site in the boundary. To make the analysis simpler, we assume that $\boldsymbol{g}^\mathcal{R}(\omega)$ posses well resolved poles, that is $\gamma \ll |\Omega_n - \Omega_m| \, \forall n \neq m$, which is always the case for a sufficiently small region $\mathcal{R}$—band edges, which have a large density of states impose a stronger condition on $\gamma$. We look for finite solutions for $\bar{\phi}_n$ under the condition that $\bar{\phi}_\alpha \sim \gamma/J \to 0$ (this defines the maximum localization/confinement). Hence, it is clear that the driving frequency must match one of the isolated poles of $\boldsymbol{g}^\mathcal{R}(\omega)$, say

$$\omega_d = \Omega_m, \tag{15}$$

in which case only the $m$-th mode is excited (describe by the normalized mode amplitude $\varphi_j^{(m)} = \langle j|\varphi_m\rangle$).

Once the frequency of the driving is fixed, we still need to find the appropriate conditions for the amplitudes $F_\beta$ to guarantee that $\bar{\phi}_\alpha \to 0$. In order to do that, we first notice

that $G_{\alpha\beta}(\omega) = \tilde{g}_{\alpha\beta}(\omega) + \tilde{g}_{\alpha\delta}(\omega)\Sigma_{\delta\rho}(\omega)G_{\rho\beta}(\omega)$ with $\tilde{g}_{\alpha\beta}(\omega)$ the surface Green function associated to the $M$ driven sites (that is, excluding (only) the region $\mathcal{R}$ from the lattice) and $\Sigma_{\alpha\beta}(\omega) = V_{\alpha n}g_{nn}^{\mathcal{R}}(\omega)V_{n\beta}$ the self-energy due to $\mathcal{R}$. In particular, $\Sigma_{\alpha\beta}(\tilde{\omega}_d) \simeq -\mathrm{i}(2/\gamma)V_{\alpha m}V_{m\beta}$ (without the implicit sum on $m$) is an anti-hermitian matrix with a single nonzero eigenvalue given by $\Sigma_{\lambda\lambda} = -\mathrm{i}(2/\gamma)\sum_{\alpha}|V_{\alpha m}|^2 = \mathrm{Tr}(\Sigma(\tilde{\omega}_d))$. On its eigenvector basis, labelled by $\tilde{\alpha}$ and $\tilde{\beta}$, where the index $\tilde{\alpha} = \lambda$ corresponds to the singular eigenvector $\boldsymbol{\lambda}$ with eigenvalue $\Sigma_{\lambda\lambda} \neq 0$, we can write

$$G_{\tilde{\alpha}\tilde{\beta}}(\tilde{\omega}_d) = \tilde{g}_{\tilde{\alpha}\tilde{\beta}}(\tilde{\omega}_d) + \Sigma_{\lambda\lambda}\frac{\tilde{g}_{\tilde{\alpha}\lambda}(\tilde{\omega}_d)\tilde{g}_{\lambda\tilde{\beta}}(\tilde{\omega}_d)}{1 - \tilde{g}_{\lambda\lambda}(\tilde{\omega}_d)\Sigma_{\lambda\lambda}}, \tag{16}$$

from where we see that only the elements $G_{\lambda\tilde{\beta}}$ and $G_{\tilde{\alpha}\lambda}$ are proportional to $\gamma$ and so the smallest—here we assume that $\tilde{\boldsymbol{g}}(\tilde{\omega}_d)$ has no singularities or edge modes. Hence, if we choose $\boldsymbol{F}$ to be proportional to the singular eigenvector, that is $\boldsymbol{F} = \xi\boldsymbol{\lambda}$ or, in terms of components,

$$F_{\beta} = \xi\frac{V_{\beta m}}{(\sum_{\alpha}|V_{\alpha m}|^2)^{1/2}}, \tag{17}$$

we obtain

$$\begin{aligned}
\bar{\phi}_{\tilde{\alpha}} &= \frac{\boldsymbol{F}\cdot\boldsymbol{\lambda}^*\,\tilde{g}_{\tilde{\alpha}\lambda}(\tilde{\omega}_d)}{1 - \tilde{g}_{\lambda\lambda}(\tilde{\omega}_d)\Sigma_{\lambda\lambda}}, \\
&\simeq \frac{-\mathrm{i}\gamma\boldsymbol{F}\cdot\boldsymbol{\lambda}^*}{2\sum_{\alpha}|V_{\alpha m}|^2}\frac{\tilde{g}_{\tilde{\alpha}\lambda}(\tilde{\omega}_d)}{\tilde{g}_{\lambda\lambda}(\tilde{\omega}_d)},
\end{aligned} \tag{18}$$

where we assumed $|\tilde{g}_{\lambda\lambda}(\tilde{\omega}_d)\Sigma_{\lambda\lambda}| \gg 1$. $\bar{\phi}_{\alpha}$ can be easily obtained after a change of basis. From Eq. (14) we then get

$$\begin{aligned}
\bar{\phi}_j &\simeq -\frac{\boldsymbol{F}\cdot\boldsymbol{\lambda}^*\sum_{\alpha}V_{m\alpha}\tilde{g}_{\alpha\lambda}(\tilde{\omega}_d)}{\tilde{g}_{\lambda\lambda}(\tilde{\omega}_d)\sum_{\alpha}|V_{\alpha m}|^2}\varphi_j^{(m)}, \tag{19} \\
&\simeq -\frac{\boldsymbol{F}\cdot\boldsymbol{\lambda}^*}{(\sum_{\alpha}|V_{\alpha m}|^2)^{1/2}}\varphi_j^{(m)}, \tag{20}
\end{aligned}$$

where, as one might expect, the spatial profile in the region $\mathcal{R}$ is dictated by the selected eigenstate $\varphi_j^{(m)}$. We notice that the strict absence of $\gamma$ in Eq. (20) is a consequence of the limit $\gamma \to 0$ we used to get some simple closed expressions. The full solution contains also some attenuation due to the decay of $g_{ij}^{\mathcal{R}}(\tilde{\omega}_d)$ with the distance between sites $i$ and $j$, that arise from the contribution of the neglected modes $n \neq m$.

In the particular case of an arbitrary 1D array, driven on sites 0 and $N$ and coupled to the inner region by a coupling $J$, Eq. (20) can be further simplified to give

$$\begin{aligned}
\bar{\phi}_j &\simeq -\frac{\boldsymbol{F}\cdot\boldsymbol{\lambda}^*}{J}\frac{\varphi_j^{(m)}}{\sqrt{|\varphi_1^{(m)}|^2 + |\varphi_{N-1}^{(m)}|^2}}, \\
&\simeq -\frac{F_0}{J}\frac{\varphi_j^{(m)}}{\varphi_1^{(m)}} = -\frac{F_0}{J_{\mathrm{eff}}^{(m)}}\varphi_j^{(m)}. \tag{21}
\end{aligned}$$

Note that the amplitude of $\bar{\phi}_1 = -F_0/J$ is fixed by the driving. Similarly $\bar{\phi}_{N-1} = -F_N/J$. Here, $J_{\mathrm{eff}}^{(m)}$ is the effective coupling between the driven sites and the eigenmode $|\varphi_m\rangle$. The homogeneous linear array discussed in the previous section is obtained from here by using: $\varphi_j^{(m)} = \sqrt{2/N}\sin(k_m j)$, $\boldsymbol{F}\cdot\boldsymbol{\lambda}^* = \sqrt{2}F_0$ and $k_m = m\pi/N$ with $m = 1,\ldots,N-1$.

The results presented in this section provide a recipe (Eqs. (15) and (17)) for the design of localized polariton fields on arbitrary lattices and might then serve as a guide for future experiments. Alternatively, one could use particular lattices with directional propagation at given frequencies to generate different patterns as discussed in Ref. [26].

# 4 Nonlinear regime

When interactions are taken into account Eq. (1) becomes nonlinear as the site energy changes $\hbar\omega_0 \mapsto \hbar\omega_0 + \hbar U|\phi_j|^2$. Nevertheless, Eq. (3) remains valid as far as the driving has a single Fourier component and provided a constant amplitude solution is possible [27]. In that case, Eq. (3) must be understood as a self-consistent equation for $\bar{\phi}$. Namely,

$$\bar{\phi} = G(\tilde{\omega}_d, \bar{\phi}) \cdot F. \tag{22}$$

There are a number of numerical protocols to find solutions for such fixed point equation. Their implementation, stability and computation requirements depend very much on the geometry and dimensionality of the lattice and the distribution of the driving sites. The numerical solutions can be very complex and present hysteric behavior—the stable/unstable solutions will depend on the way the driving amplitude is set up (ramping up or down). For the same reasons, analytical solutions or approximations are in general rather difficult to find. It should be noted though, that some of the formal relations found for the linear case, such as Eqs. (4), (13) and (14), are still valid (as self-consistent equations) as well as some of the approximations that followed them. Of course, one can still argue that if the condition for localization is met ($|\bar{\phi}_\alpha| \propto \gamma/|J| \ll 1$) then $\hbar\omega_d$ should match an eigenenergy ($\hbar\bar{\Omega}_m$) of the self-consistent Hamiltonian $H_\mathcal{R}$—provided its eigenenergies are well resolved. Of course, since now $\varphi_j^{(m)}$ is self-determined, so is $\lambda$ and one cannot longer guarantee that $F$ (externally fixed) is proportional to it. Nevertheless, the analytical results of the previous section might serve as a guide to interpret the fully numerical results. There are, however, special cases that admit some treatment as the 1D systems we now discuss.

## 4.1 Nonlinear linear array

Let us consider the nonlinear version of the linear array discussed in Sec. 3.1. Figure 3 shows the numerical results obtained using Eq. (22) for $N = 10$ and for four different values of $U/|J| = 0, 0.1, 0.2, 0.5$ with even driving ($F_0 = F_{10}$). The 'internal' region $\mathcal{R}$ is then defined by sites 1 to 9. The interaction term is included only from sites $-9$ through 19, a justified approximation when looking at states localized inside $\mathcal{R}$, that allows us to include the rest of the (infinite) sites as a self-energy on the boundary sites ($-9$ and 19). The self-consistent procedure is initiated with $\bar{\phi} = 0$ and it is carried out until $|\Delta\bar{\phi}|/|\bar{\phi}| < 10^{-6}$. There are several important aspects of Fig. 3 to point out: (i) the colormaps of $|\bar{\phi}_j|^2$ (left panels) shows that in all cases the resonantly driven localization persist for the appropriate value of $\omega_d$; (ii) the interaction blue shifts the resonant condition, as it is clearly evident on the right panels where we plot $|\bar{\phi}_5|^2$ (center of $\mathcal{R}$) and $|\bar{\phi}_{-5}|^2$ (outside $\mathcal{R}$), see discussion below; (iii) the extend of the localization is not substantially affected by $U$ in the range presented here; (iv) the noisy data near the band edges reflect the presence of instabilities where convergence is not possible.

To better understand these numerical results, we can take advantage of the analytical expressions found before. In this case one has that $\varphi_1^{(m)} = \pm\varphi_{N-1}^{(m)}$, that is, the even/odd symmetry is preserved and so the choice $F_0 = \pm F_N$ guarantees that $F \parallel \lambda$. Therefore, one can make use of Eq. (21) to express the local interaction potential in terms of the self-consistent mode,

$$U|\bar{\phi}_j|^2 = \frac{U|F_0|^2}{|\varphi_1^{(m)}|^2 J^2}|\varphi_j^{(m)}|^2 = \frac{U_{\text{eff}}^{(m)}|F_0|^2}{J^2}|\varphi_j^{(m)}|^2, \tag{23}$$

where we introduced an effective $U_{\text{eff}}^{(m)} > U$ instead of $J_{\text{eff}}^{(m)}$. This leads to an interesting and non-trivial consistency problem whose full solution is beyond the scope of the present work. Here we only consider some limiting scenarios. First, we notice that $U_{\text{eff}}^{(m)}$ near the band center

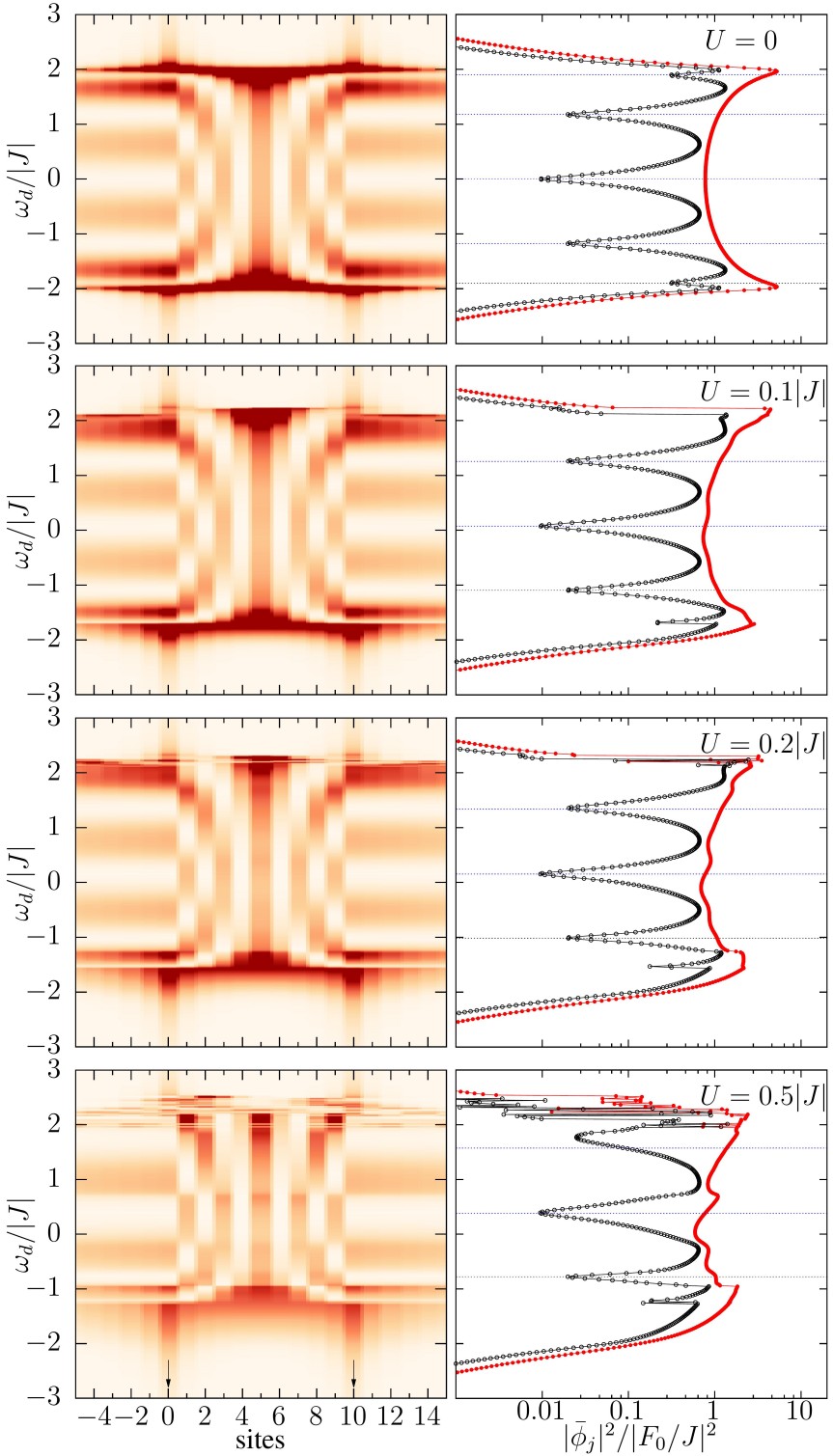

Figure 3: $|\bar{\phi}_j|^2$ as a function of the driving frequency $\omega_d$ for the interacting linear array described in the text with $N = 10$ (sites 0 and 10 are driven, indicated by the black arrows). Left panels shown a colormaps for sites $-5$ to 15 while in the right panels only the central site amplitude, $|\bar{\phi}_5|^2$ (solid symbols), and $|\bar{\phi}_{-5}|^2$ (open symbols) are shown. The dashed horizontal lines in the later correspond to the estimated shifts, Eq. (24), including the factor $(1-2\mu_m)$, see text. Here $U/|J| = 0$, 0.1, 0.2 and 0.5, from top to bottom, and $\gamma/|J| = 0.05$.

is smaller than towards the band edges, so one expect the interaction effects to be stronger in the latter case. In the limit $U/|J| \ll 1$ we can applied first order perturbation theory to calculate the shift of the localization condition due to the interaction, namely

$$
\begin{aligned}
\omega_d &= \bar{\Omega}_m, \\
\omega_d &\simeq \Omega_m + \frac{U_{\text{eff}}^{(m)}|F_0|^2}{J^2} \sum_{j=1}^{N-1} |\varphi_j^{(m)}|^4, \\
&\simeq \Omega_m + \frac{2}{N} \frac{U|F_0|^2}{J^2 \sin^2(k_m)} \sum_{j=1}^{N-1} \sin^4(k_m j).
\end{aligned}
\tag{24}
$$

When $N$ is even, which corresponds to an odd number of sites between the driven sites, the state in the band center has $k_m = \pi/2$, given $\omega_d = \omega_0 + U|F_0|^2/J^2$ as the condition for optimal localization (this agrees with the numerical results obtained in [24] for $N = 2$ and $N = 6$). In fact, this is a result valid beyond perturbation theory as $\varphi_j^{(N/2)} = \sqrt{2/N} \sin(j\pi/2)$ is an exact self-consistent eigenfunction of $H_{\mathcal{R}}$ in the presence of an uniform energy shift—though not necessarily stable for large $U$. This is so because in that mode all sites in $\mathcal{R}$ have zero or equal amplitude. We note however, that this is valid under the simplifications used to derive Eq. (21).

In order to compare the shift predicted by Eq. (24) with the numerical results, and since the latter where obtained using a small but finite value of $\gamma$, it is important to retain in Eqs. (18) and (20) corrections up to order $(\gamma/J)^2$ and $\gamma/|J|$, respectively. This adds a factor $(1-\mu_m)$ to Eq. (23) and $(1-2\mu_m)$ to the shift in Eq. (24),

$$
\begin{aligned}
\delta\omega_d &= \omega_d - \Omega_m, \\
&\simeq \frac{2}{N} \frac{U|F_0|^2(1-2\mu_m)}{J^2 \sin^2(k_m)} \sum_{j=1}^{N-1} \sin^4(k_m j).
\end{aligned}
\tag{25}
$$

with $\mu_m = \gamma N/(4|J|\sin(k_m))$. This renormalized frequency shift is indicated in Fig. 3 with dashed lines, showing a remarkable agreement despite the fact that $U/|J|$ is not necessarily small. A similar comparison, and agreement, is presented in Fig. 4 for $N = 2$ up to a even higher value of $U/|J|$. Here the shift is given by

$$
\omega_d \simeq \omega_0 + \frac{U|F_0|^2}{J^2}\left(1 - \frac{\gamma}{|J|}\right),
\tag{26}
$$

and it is indicated in panel (a) for each value of $U$ by the corresponding arrows. In panel (b) we subtracted the shift to show that there is a small enhancement of the localization (width of minimum of $|\bar{\phi}_{-1}|^2$) with increasing $U$. This shift induced by the interactions serves as another knob to tune the localization as now both $\omega_d$ and $F_0$ determine the optimal condition [24].

## 4.2 Nonlinear SSH chain

Another simple but experimentally relevant case [20] corresponds to the SSH chain where two consecutive sites, say 1 and 2, are driven. We assume they are linked by the strongest coupling $J$, though some of the following results are independent of that. Note that here there is no 'internal' region $\mathcal{R}$ so some of the previous results do not directly apply.

The two driven sites constitute an effective molecule embedded on a lattice. The exact effective Green function for the driven sites can be cast as,

$$
\begin{aligned}
\tilde{G}(\omega) &= \begin{pmatrix} \tilde{G}_{11}(\omega) & \tilde{G}_{12}(\omega) \\ \tilde{G}_{21}(\omega) & \tilde{G}_{22}(\omega) \end{pmatrix}, \\
&= \frac{1}{\mathcal{D}(\omega)} \begin{pmatrix} \omega - \omega_2 - \Sigma_R(\omega) & J \\ J & \omega - \omega_1 - \Sigma_L(\omega) \end{pmatrix},
\end{aligned}
\tag{27}
$$

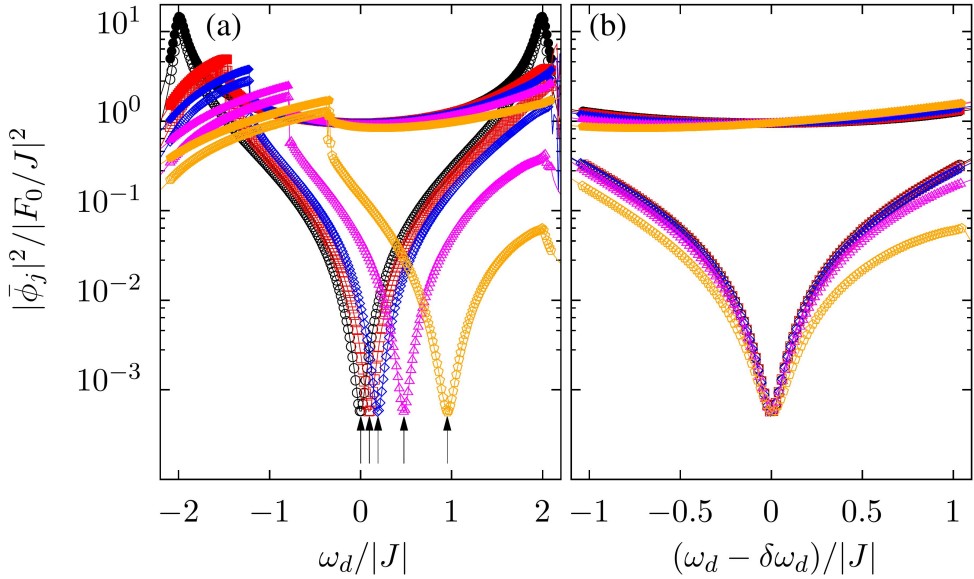

Figure 4: (a) Plot of $|\bar{\phi}_1|^2$ (solid symbols) and $|\bar{\phi}_{-1}|^2$ (open symbols) as a function of the driving frequency $\omega_d$ for the linear array described in the text with $N = 2$ (sites 0 and 2 are driven). Here $U/|J| = 0$ (circles), 0.1 (squares), 0.2 (diamonds), 0.5 (triangles) and 1 (pentagons). The arrows indicate the corresponding predictions of Eq. (26). Here we took $\omega_0 = 0$ and $\gamma/|J| = 0.05$. (b) Same as before but with the predicted shift $\delta\omega_d$ subtracted.

where $\omega_{1,2} = \omega_0 + U|\bar{\phi}_{1,2}|^2$, $\mathcal{D}(\omega) = \mathrm{Det}(\tilde{G}(\omega))^{-1}$ and $\hbar\Sigma_{L,R}(\omega)$ are the boundary self-energies due to the semi-infinite chain attached to each of the driven sites ($\Sigma_L$ to site 1 and $\Sigma_R$ to site 2). They can be written as $\Sigma_L(\omega) = J'^2 g_{33}(\omega)$ and $\Sigma_R(\omega) = J'^2 g_{00}(\omega)$ where $g_{33}(\omega)$ and $g_{00}(\omega)$ are the local surface Green functions of the respective semi-infinite chain, which do not explicitly depend on $\bar{\phi}_1$ and $\bar{\phi}_2$. The later are given by,

$$
\begin{aligned}
\bar{\phi}_1 &= \tilde{G}_{11}(\tilde{\omega}_d)F_1 + \tilde{G}_{12}(\tilde{\omega}_d)F_2, \\
\bar{\phi}_2 &= \tilde{G}_{21}(\tilde{\omega}_d)F_1 + \tilde{G}_{22}(\tilde{\omega}_d)F_2.
\end{aligned}
\tag{28}
$$

*Symmetric solution–.* We start by searching for symmetric solutions with respect to the two driven sites for the case $F_1 = F_2 = F$. In that case, $\tilde{G}_{11} = \tilde{G}_{22}$ and $\bar{\phi}_1 = \bar{\phi}_2 = \bar{\phi}$ so that Eq. (28) reduces to

$$
\bar{\phi} = \frac{F}{\delta\omega' + i\gamma'/2 - U|\bar{\phi}|^2 - J},
\tag{29}
$$

with $\delta\omega' = \delta\omega - \mathrm{Re}[\Sigma(\tilde{\omega}_d)]$, $\delta\omega = \omega_d - \omega_0$, and $\gamma' = \gamma - 2\mathrm{Im}[\Sigma(\tilde{\omega}_d)]$. Eq. (29) has the form of the standard bistability equation for a driven non-linear resonator [27], except for the fact that $\Sigma(\omega)$ depends on the amplitude of the other sites through the local nonlinear terms. To proceed, we assume that $\omega_d$ lies inside the SSH gap, that is $|\delta\omega| < |J - J'|$. We then expect that the amplitudes $\bar{\phi}_j$ will be strongly localized around the driven sites. If the localization is strong enough, as a first approximation, one can ignore the nonlinear terms on the non-driven sites so that $\Sigma(\omega)$ has a closed analytical form (not shown). It can be easily checked that for $\delta\omega = 0$ the exact self-energy is purely imaginary, while near the center of the gap it takes the form

$$
\Sigma(\tilde{\omega}_d) \sim -\left(\delta\omega + i\frac{\gamma}{2}\right)\frac{J'^2}{J^2 - J'^2}.
\tag{30}
$$

where $|\delta\omega|, \gamma \ll |J - J'|$ and $|J| > |J'|$. It is then clear that Eq. (29) corresponds to an effectively isolated dimer with slightly renormalized parameters, the only role of the rest of

the SSH chain in determining the stationary values of $\bar{\phi}$. Specifically, $\delta\omega' = \epsilon\delta\omega$ and $\gamma' = \epsilon\gamma$, with $\epsilon = J^2/(J^2 - J'^2)$. A standard stability analysis of Eq. (29) yields two unstable thresholds, up to corrections of order $\gamma'^2/|J|$,

$$
\begin{aligned}
(U|\bar{\phi}_+|^2, F_+) &= \left( \frac{-J + \delta\omega'}{3}, \frac{2}{3\sqrt{3}} \sqrt{\frac{(-J + \delta\omega')^3}{U}} \right), \\
(U|\bar{\phi}_-|^2, F_-) &= \left( -J + \delta\omega', \frac{\gamma'}{2} \sqrt{\frac{-J + \delta\omega'}{U}} \right),
\end{aligned}
\tag{31}
$$

that correspond to the situation where $F$ is increased ($+$) or decreased ($-$). Notice that the sign of $J$, which we take to be negative, is important for the symmetric solution to be possible (when $J > 0$ the symmetric solution requires $F_1 = -F_2$).

The exact numerical solution of Eq. (22) for an infinite lattice but where interactions are included only on seven unit cells (14 sites), three on each side of the one being driven, is shown in Fig. 5. Panel (a) shows a colormap of $|\bar{\phi}_j|^2$ as a function of $F$ (ramped up) while panel (b) shows the behavior of $|\bar{\phi}_2|^2$, $|\bar{\phi}_3|^2$ and $|\bar{\phi}_4|^2$. The alternating low power spatial profile can be understood, as discussed in the previous cases, in terms of the surface lattice Green function in the absence interactions. Note that there is an excellent agreement with the threshold estimated by Eq. (31) as indicated in the figure by the pentagon symbol at the lowest value of $F/F_{\text{th}}$.

Figure 5(a-b) also shows, in agreement with the experimental data of Ref. [20], a second transition at a higher power. This is a signal that the assumption of negligible interaction effects for the undriven sites is not longer valid. In fact, it is clear from the figure that neighboring sites acquire a significant occupation after the first transition. To take this into account we can use the fact that $\bar{\phi}_j = g_{j3}(\tilde{\omega}_d)J'\bar{\phi}$ with $j \geq 3$ (cf. Eq. (13)). If interactions are only included in the two neighboring sites, 3 and 4 ( sites 0 and $-1$ are taken into account by the assumed symmetry of the solution), this can be viewed as an additional effective molecule 'driven' by the field $J'\bar{\phi}$ but with a SSH chain attached to site 4 only. Following a similar procedure as before, for the case $\delta\omega = 0$ and for increasing power, one finds that the second transition takes place when $U|\bar{\phi}_2|^2 = -(64/81\sqrt{3})J^3/J'^2$, $U|\bar{\phi}_3|^2 = -J/3\sqrt{3}$, and $U|\bar{\phi}_4|^2 = -J/\sqrt{3}$. The threshold value for $F$ can be obtained from Eq. (29) taking into account that the contribution to $\delta\omega'$ from the self-energy is the shift $(3\sqrt{3}/8)J'^2/J$—once again, this is the only effect of the rest of the SSH chain. These estimates, also shown in Fig. 5(b), are in very good agreement with the full numerical data. Note that the blue shift of the driven sites can be quite significant as $|J/J'| > 1$.

*Asymmetric solution–.* We now look for the conditions to obtain an asymmetric solution where only the sites on one side of the driven sites are significantly populated. For that, we impose that one of the driven sites, say site 1, has zero (or very small) amplitude—this immediately implies that $\bar{\phi}_j = g_{j0}(\tilde{\omega}_d)J'\bar{\phi}_1$ with $j \leq 0$ are also zero (very small) as $g_{j0}(\tilde{\omega}_d)$ has no poles (here we assume $|J| > |J'|$ as before and $\omega_d$ to be inside the SSH gap). Making $\bar{\phi}_1 = 0$ in Eq. (28) leads to

$$
\begin{aligned}
\bar{\phi}_2 &= \left( \tilde{G}_{21}(\tilde{\omega}_d) - \tilde{G}_{22}(\tilde{\omega}_d)\frac{\tilde{G}_{11}(\tilde{\omega}_d)}{\tilde{G}_{12}(\tilde{\omega}_d)} \right) F_1, \\
&= -\frac{\text{Det}(\tilde{\boldsymbol{G}}(\tilde{\omega}_d))}{\tilde{G}_{12}(\tilde{\omega}_d)} F_1 = -\frac{F_1}{J},
\end{aligned}
\tag{32}
$$

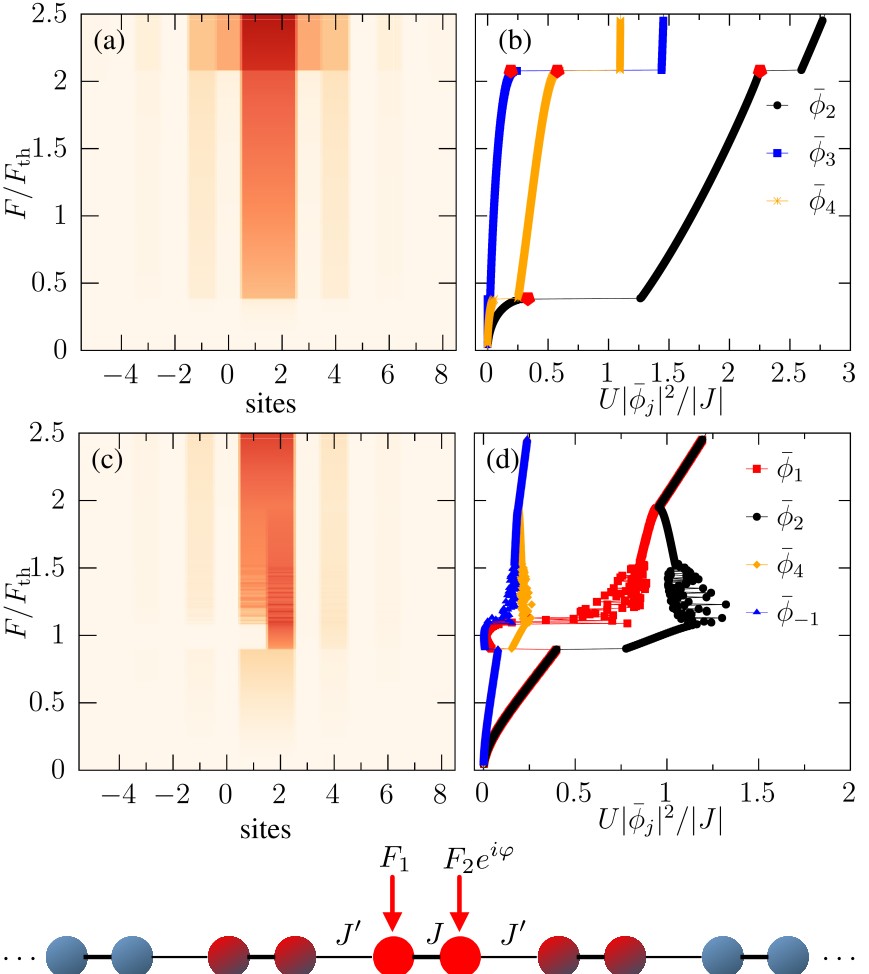

Figure 5: Left panels: colormap of $|\bar{\phi}_j|^2$ as a function of the driving field amplitude $F = |F_1| = |F_2|$ (ramped up) for the SSH chain (see scheme at the bottom) for two different relative phases $\varphi = 0$ (a) and $\varphi = \pi$ (c). Other parameters are: $J' = 0.45J$, $U = 0.1|J|, \gamma = 0.1|J|$, $F_{th}^2 = |J|^3/U$, $\delta\omega = 0$ and $J < 0$. Right panels: corresponding amplitudes for some relevant sites. Red pentagons in (b) indicate transition points obtained from Eq. (31).

which is valid even in the presence of non-linear terms in all sites. In addition,

$$\begin{aligned} \frac{F_2}{F_1} &= -\frac{\tilde{G}_{11}(\tilde{\omega}_d)}{\tilde{G}_{12}(\tilde{\omega}_d)}, \\ &= \frac{\delta\omega' + i\gamma'/2 - U|\bar{\phi}_2|^2}{-J}. \end{aligned} \tag{33}$$

This imposes a precise requirement for the phase relation between the two driving lasers, as in the polariton molecule discussed before. For small detuning, $|\delta\omega'| < U|\bar{\phi}_2|^2$, as it is the case considered here, this requires a phase difference $\varphi \simeq \pi$ (0) for $J < 0$ ($J > 0$). If both lasers have the same or similar amplitudes, $|F_2/F_1| \simeq 1$, that is the experimental situation in Ref. [20], Eq. (33) cannot be satisfied exactly, which also means that $\bar{\phi}_1 \neq 0$. Nevertheless, since $\gamma'/|J| \ll 1$, and if we take $F_2 = -F_1$, then $|\bar{\phi}_1|$ gets minimized (to the lowest order in $\gamma'/|J|$) when the real part of Eq. (33) is $-1$. That is,

$$U|\bar{\phi}_2|^2 = -J + \delta\omega', \qquad |F_1|^2 = F_{th}^2 = (-J + \delta\omega')J^2/U, \tag{34}$$

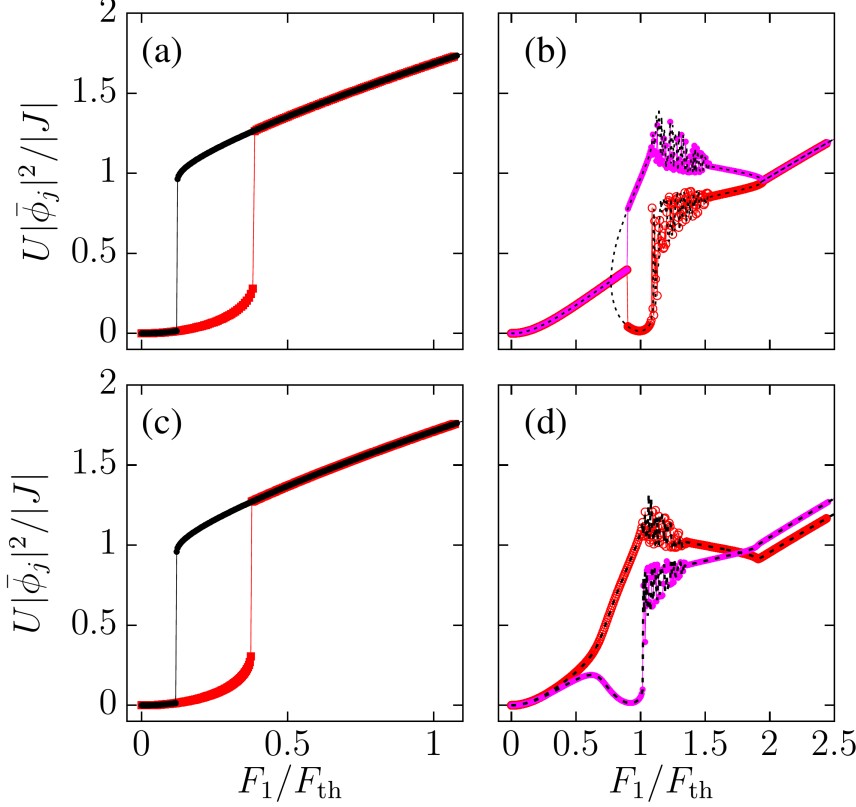

Figure 6: Low power hysteresis loop for the SSH chain. Only the amplitudes of the two driven sites are shown. Red/magenta (black) symbols/lines correspond to increasing (decreasing) driving amplitude. On the top panels we use $|F_2| = |F_1|$ while in the bottom ones $|F_2| = 1.05|F_1|$. Panels (a) and (c), where $\varphi = 0$, show a standard hysteresis loop. Panels (b) and (d), where $\varphi = \pi$, show a more complex behavior with dynamical instabilities (noisy data) in both cases but only hysteric behavior on (b). Open and closed symbols correspond to sites 1 and 2, respectively. The rest of the parameters are: $\omega_d = \omega_0$, $U = 0.1|J|$, $\gamma = 0.1|J|$, $J' = 0.45J$ and $J < 0$.

which implies $|\bar{\phi}_1| \sim (\gamma'/2)F_{\mathrm{th}}/J^2$. The amplitude of the sites to the right of site 2 follow the spatial dependence dictated by $\bar{\phi}_j = g_{j3}(\tilde{\omega}_d)J'\bar{\phi}_2$ with $j \geq 3$, which has a large amplitude only on the same sublattice sites. This explains the experimental results of Ref. [20]. The numerical results for this case are presented in Figs. 5 (c) and (d) where a strong asymmetric solution develops for $|F_1|^2 = -J^3/U$—here $\omega_d = \omega_0$. The noisy data in Figs. 5(d) reflects the presence of dynamical instabilities.

*Hysteresis–.*Figure 6 presents the low power hysteresis loop for both the symmetric $\varphi = 0$ [(a)-(c)] and anti-symmetric $\varphi = \pi$ [(b)-(d)] cases obtained from the numerical solution of Eq. (22). On the top panel both driven sites have the same driving field amplitude, $|F_1| = |F_2|$, while in the bottom ones a small asymmetry was introduced, $|F_2| = 1.05|F_1|$, both to mimics more realistic scenarios and to favor the spontaneous spatial symmetry breaking. The symmetric solutions shows always an hysteric behavior with thresholds that are in agreement with Eq. (31). The asymmetric case, on the other hand, is more involved probably due to the singular behavior of the surface Green function due to the SSH edge state. In particular, hysteric behavior is only found for $|F_1| = |F_2|$ (panel (b)). Noisy data signaling the presence of dynamical instabilities is always present after the transition near $F_{\mathrm{th}}$.

Figure 7 shows the dependence on the relative phase of the laser, that determines the side of the chain that is populated near $\varphi = \pi$ (symmetry breaking point), for two different values

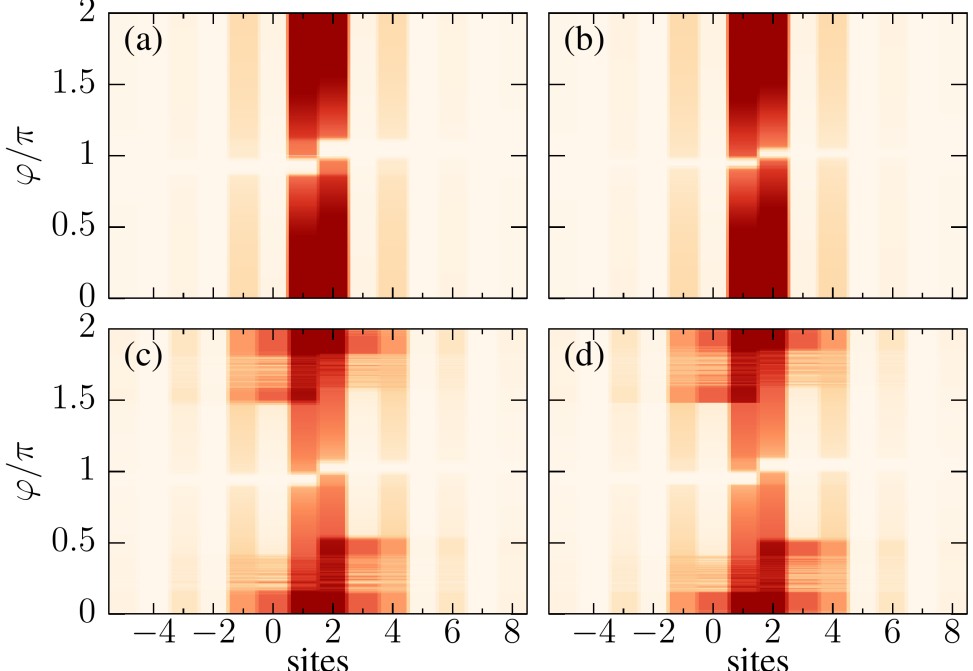

Figure 7: Phase dependence of a two-sites driven SSH chain (see scheme of Fig. 5) for $\omega_d = \omega_0$, $U = 0.1|J|$, $\gamma = 0.1|J|$ and $J < 0$. Top panels: interactions are included in 14 sites, $J' = 0.45J$, $|F_2| = 1.02|F_1|$; (a) $F_1 = 0.96F_{\text{th}}$; (b) $F_1 = 1.04F_{\text{th}}$. Bottom panels: interactions are included in 18 sites, $J' = 0.55J$, $F_1 = F_{\text{th}}$; (c) $|F_2| = 1.02|F_1|$; (d) $|F_2| = (1 + \delta f)|F_1|$ with $\delta f \in [-0.02, 0.02]$ a random variable. In (a), (b) and (c) the data was averaged over ten realizations of an added small random phase fluctuation $\delta\varphi \in [-\pi/50, \pi/50]$.

of the SSH gap—the later controls the extend of the edge mode and hence the relevance of the interaction on the neighboring sites. The self-consistency iteration is started with $\bar{\phi} = 0$. The top panel correspond to the case of a larger SSH gap and hence a stronger localization of the soliton mode. To test the resilience of the effect and mimic possible experimental effect we added a small fluctuation both in phase and amplitude of the driving field as indicated in the caption. In all cases the emergence of the asymmetric soliton mode near $\varphi = \pi$ is apparent from the figure. It is worth pointing out the resemblance of our numerics with the experimental data of Ref. [20], as for instance the appearance of asymmetric thresholds at higher phase differences.

## 5   Summary and conclusions

Using lattice Green function techniques we have analyzed how resonant driving engineering can be used in polariton or photonic arrays to tailor spatially localized states.

In the linear regime, our approach is exact and allowed us to determine the precise conditions on the laser field to obtain the maximum localization of the polariton field on the region delimited by the driving lasers. We believe this would be very helpful both to design particular driving setups and, hopefully, to better understand their experimental outcomes. Generalizations to the case of spatially periodic driving is straightforward. Additionally, a similar scheme as the one described in Ref. [26] could be also used in the present context to generate different patterns on particular lattices. In another interesting direction, slow time modulations of

the driving field, either the phase and/or the amplitude, could also be addressed within this method by using appropriate adiabatic expansions of the Green function, hence allowing to properly account for the presence of topological pumping effects—work on this direction is underway and will be presented elsewhere.

The treatment of nonlinear effects, on the other hand, is in general more involved as they usually lead to some complex dynamical behaviour with the presence of hysteresis loops and instabilities. Yet, the approach presented here, when thought as a self-consistent fixed point equation, allows a clear understanding of several features of the solutions and their experimental implementations. In fact, for simple 1D cases, we were able to obtain analytical expressions for the energy shifts, mode amplitudes and power thresholds that agree quit well with the full numerics. In more complicated situations or in higher dimensions, we expect that our approach will be also helpful, providing theoretical insights and/or appropriate initial configurations for full time dependent calculations.

*Note*: Upon completion of this work we learned of Ref. [24] where localization in polariton lattices in the presence of optical Kerr non-linearities was studied with some overlapping results for the linear array case.

# Acknowledgements

We thank Alberto Amo for fruitful discussions and Jacques Tempere, Michiel Wouters, and Nathan Goldman for their hospitality at UAntwerp and ULB, respectively.

**Funding information** We acknowledge financial support from the ANPCyT-FONCyT (Argentina) under grants PICT 2018-1509 and PICT 2019-0371, SeCyT-UNCuyo grant 06/C053-T1 and F.R.S.-FNRS research stay grant.

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
