# Peer review of "Localization engineering by resonant driving in dissipative polariton arrays"

_SciPost Physics Core, doi:SciPost Phys. Core 7, 052 (2024)_

## Round 1 · Referee Report · Alex Ferrier (Referee 1) · 2024-7-22

Strengths
1- Does a good job of situating the work as relevant within the context of recent developments in the field, and noting throughout where results of the theory being presented correspond with experimental observations in the references.
2- Describes both the most general application of the theory to arbitrary geometries, as well as specific cases of common interest in the field such as the SSH model.
3- Is clearly and logically structured, demonstrating how the method works for the simplest cases with one or two driven sites before progressing to arbitrary geometries and nonlinear interactions.
Weaknesses
1- There are some places where the formatting of equations or figures needs to updated to fit properly in the style and layout of SciPost.
2- Some of the figures could be adjusted for improved readability.
3- While the manuscript does clearly explain how the method can be used for arbitrary lattice geometries, results are only presented for 1D examples.
Report
The complaints I have about the manuscript in its current form are mostly matters of presentation. There a few instances where it looks like the manuscript has not been properly updated to account for the formatting in SciPost: for instance, there is an ambiguous construction $(H)_{ij} = H_{ij}/\hbar$ on line 79 and similarly on line 191; I noticed in the ArXiv version these are properly distinguished by font, but in SciPost all equation text is bold so the two quantities $H$ need to be distinguished from each other in some other way. There is also figure 3, whose dimensions are currently such that the caption is clipping into the page number. I noticed a few cases where it looked like some of the axes labels or captions of figures had not been properly updated from some older version; I will include a full list of corrections I noticed in the "Requested changes" below. I also thought that some of the figures could be adjusted slightly to improve readability: in particular, often the density of points is such that it can be difficult to distinguish "open symbols" and "solid symbols", especially in panels b) and d) of figure 6 where they are only really distinguishable in the small region of dynamical instabilities; perhaps since the data points look almost continuous solid/dashed lines could be used instead of solid/open symbols.
Finally, I had a few questions for the author that I was curious about, which they may also want to further elaborate on in the manuscript if they so choose. First, since all the results presented in the manuscript are for 1D models, I was wondering what the author's assessment was of how difficult it might be in practice to apply the construction in section 3.2 to the sort of general 2D case implied by the diagram in figure 2? Second, I had some questions about the element of random fluctuations in figure 7. Is there any particular reason for the size of fluctuations being considered? Since only a small number of realisations were used, am I to understand that this was enough to determine that any fluctuations of this size have a negligible effect on the results? Was each realisation of the fluctuations fixed as $\varphi$ is varied or was it randomised separately at each value of $\varphi$? Does the author have any estimate or intuition for how large these fluctuations would need to be to disrupt the effect?
Overall, I think that this work has an important place in its field, particularly as a tool to assist in the design of future theoretical and experimental studies of these systems, and am happy to recommend it for publication in SciPost Physics Core, provided that certain necessary corrections to the presentation can be implemented.
Requested changes
1- Different quantities labelled $H$ (with or without factor of $\hbar$) need to be properly distinguished from each other around line 79 and again around line 191.
2- The size or arrangement of figure 3 should be adjusted to that the caption does not overlap the page number.
3- Caption of figure 3 refers to "white arrows" which are now actually black arrows. In my opinion, figure 1 might also be improved by adding similar arrows to more strongly indicate the driven sites.
4- While it is fairly clear what is meant by "hopping matrix", it might be even better if $V$ introduced in section 3.2 is defined unambiguously in terms of the original model in equation 1.
5- I think the quantity $D(\omega)$ in equation 27 is improperly defined in line 303 below. Is it not supposed to be the determinant of the matrix being multiplied by $1/D(\omega)$, not of the the whole of $\tilde{G}$ including that factor? If so can the definition or equation 27 be adjusted to clarify this.
6- The x-axis labels of figure 5 b) & d), as well as the bottom y-axis label of figure 6 refer to a parameter $t$ which I assume is supposed to be $J$.
7- The way panels b) & d) of figure 6 are plotted could be adjusted so that the data for sites 1 & 2 is more easy to distinguish.
Recommendation
Ask for minor revision

---

## Round 2 · Referee Report · Alex Ferrier (Referee 1) · 2024-7-26

Report

I feel that all my comments from the previous report have been sufficiently addressed and so am happy to recommend the manuscript for publication at this stage.

I have a final very minor suggestion, insomuch as the author has an opportunity to make small corrections to the text in the production stage, that it might improve the clarity if for figures where different symbols are also distinguished by colour, the colour of the symbols is also mentioned in the caption e.g.:
- "red solid symbols"/"black open symbols" in figure 3;
- "(black circles)", "(red squares)" etc. in figure 4;
- "Red open and magenta closed symbols …" in figure 6;
especially for the case of figure 6 where these colours were added in the latest revision to help readers more easily distinguish these different symbols.

Recommendation

Publish (meets expectations and criteria for this Journal)

---

## Round 2 · Author Response

I thank the Referee for the constructive comments and for pointing out some typos which I have now corrected.

As for the Referee questions: (i) implementations in 2D are numerically more demanding but straightforward. In the linear case there is no difficulty as there are many ways to efficiently calculated the lattice Green function. For the nonlinear case, a good routine for solving fixed point equations have to be added to the Green function calculation protocol. This is feasible and, for a few simple cases I tested, it works fine; (ii) the phase fluctuation was randomized for each value of the phase with a fine grid for the later. A small number of realizations was enough to determine the average.

---

## Round 2 · List of Changes

1-The bold style of the inline equations have been removed (this was due to a \mathbold command on the abstract section of the SciPost template).
2- Comments (2-7) of the Referee were addressed as suggested.
3- Reference [24] was updated.

---

## Editorial Decision

published